# Spaceflight induces changes in gene expression profiles linked to insulin and estrogen

Begum Aydogan Mathyk [1,26✉], Marshall Tabetah [2,26], Rashid Karim [3,4,26], Victoria Zaksas [5,6,26], JangKeun Kim [7,26], R. I. Anu [8,26], Masafumi Muratani [9,10], Alexia Tasoula [11], Ruth Subhash Singh [12], Yen-Kai Chen [13], Eliah Overbey [7], Jiwoon Park [7], Henry Cope [14], Hossein Fazelinia [15], Davide Povero [16], Joseph Borg [17], Remi V. Klotz [18], Min Yu [18], Steven L. Young [19], Christopher E. Mason [7], Nathaniel Szewczyk [14,20,27], Riley M. St Clair [21,27], Fathi Karouia [22,23,27] & Afshin Beheshti [24,25,27✉]

Organismal adaptations to spaceflight have been characterized at the molecular level in model organisms, including *Drosophila* and *C. elegans*. Here, we extend molecular work to energy metabolism and sex hormone signaling in mice and humans. We found spaceflight induced changes in insulin and estrogen signaling in rodents and humans. Murine changes were most prominent in the liver, where we observed inhibition of insulin and estrogen receptor signaling with concomitant hepatic insulin resistance and steatosis. Based on the metabolic demand, metabolic pathways mediated by insulin and estrogen vary among muscles, specifically between the soleus and extensor digitorum longus. In humans, spaceflight induced changes in insulin and estrogen related genes and pathways. Pathway analysis demonstrated spaceflight induced changes in insulin resistance, estrogen signaling, stress response, and viral infection. These data strongly suggest the need for further research on the metabolic and reproductive endocrinologic effects of space travel, if we are to become a successful interplanetary species.

[1] Department of Obstetrics and Gynecology, University of South Florida Morsani College of Medicine, Tampa, FL, USA. [2] Department of Agricultural and Biological Engineering, Purdue University, West Lafayette, IN 47907, USA. [3] Department of Electrical Engineering and Computer Science, University of Cincinnati, Cincinnati, OH 45220, USA. [4] Novartis Institutes for Biomedical Research, 181 Massachusetts Ave, Cambridge, MA 02139, USA. [5] Center for Translational Data Science, University of Chicago, Chicago, IL 60637, USA. [6] Clever Research Lab, Springfield, IL 62704, USA. [7] Department of Physiology and Biophysics and World Quant Initiative for Quantitative Prediction, Weill Cornell Medicine, New York, NY 10021, USA. [8] Department of Cancer Biology & Therapeutics, Precision Oncology and Multi-omics clinic, Genetic counseling clinic. Department of Clinical Biochemistry, MVR Cancer Centre and Research Institute, Calicut, India. [9] Transborder Medical Research Center, University of Tsukuba, Ibaraki 305-8575, Japan. [10] Department of Genome Biology, Faculty of Medicine, University of Tsukuba, Ibaraki 305-8575, Japan. [11] Department of Life Science Engineering, FH Technikum, Vienna, Austria. [12] Department of Biophysics, University of Mumbai, Mumbai 400029, India. [13] School of Biological Sciences, University of Auckland, Auckland, New Zealand. [14] School of Medicine, University of Nottingham, Derby DE22 3DT, UK. [15] Department of Biomedical and Health Informatics and Proteomics Core Facility, Children's Hospital of Philadelphia, 3401 Civic Center Blvd, Philadelphia, PA 19104, USA. [16] Division of Gastroenterology and Hepatology, Mayo Clinic, Rochester, MN 55905, USA. [17] Department of Applied Biomedical Science, Faculty of Health Sciences, Msida MSD2090, Malta. [18] Department of Stem Cell Biology & Regenerative Medicine, University of Southern California, Los Angeles, CA, USA. [19] Division of Reproductive Endocrinology and Infertility, Duke School of Medicine, Durham, NC, USA. [20] Ohio Musculoskeletal and Neurological Institute, Heritage College of Osteopathic Medicine, Ohio University, Athens, OH 45701, USA. [21] Department of Life Sciences, Quest University, Squamish, BC, Canada. [22] Blue Marble Space Institute of Science, Exobiology Branch, NASA Ames Research Center, Moffett Field, CA, USA. [23] Space Research Within Reach, San Francisco, CA, USA; Center for Space Medicine, Baylor College of Medicine, Houston, TX, USA. [24] Stanley Center for Psychiatric Research, Broad Institute of MIT and Harvard, Cambridge, MA, USA. [25] Blue Marble Space Institute of Science, Space Biosciences Division, NASA Ames Research Center, Moffett Field, CA, USA. [26]These authors contributed equally: Begum Aydogan Mathyk, Marshall Tabetah, Rashid Karim, Victoria Zaksas, JangKeun Kim, R. I. Anu [27]These authors jointly supervised this work: Nathaniel Szewczyk, Riley M. St Clair, Fathi Karouia, Afshin Beheshti. ✉email: abegum@usf.edu; afshin.beheshti@nasa.gov

We have entered an era in which living and working in space is a reality. Studies show that spaceflight has health risks such as central nervous system and cardiovascular dysfunction, cancer risk, immune dysregulation, muscle and bone loss, and fatty liver[1]. At the molecular level, risks include oxidative stress, DNA damage, cosmic radiation exposure, mitochondrial dysfunction, microbiome shift, and epigenetic alterations[1]. Of these, mitochondrial dysfunction and oxidative stress have been identified as major space-related risks[2]. Mitochondria are not only the powerhouse of the cell but also the center of many metabolic pathways. Insulin is a highly conserved anabolic and mitogenic peptide hormone that acts as a master regulator of energy metabolism across species, including humans[3]. It is secreted by the pancreas in response to increased plasma glucose, and exerts its effects on most organs including the kidney and liver as well as skeletal muscle, and bone[4]. Insulin receptor is a transmembrane receptor belonging to the family of receptor tyrosine kinases (RTKs)[5] and undergoes autophosphorylation upon activation, leading to a downstream signaling cascade which influences cellular function, particularly glucose metabolism. Insulin resistance (IR) arises when these downstream molecular pathways do not respond appropriately in target tissues, thus impeding the physiologic response to insulin. In the physiological state, IR can also be observed due to opposing actions from counterregulatory hormones such as glucagon, glucocorticoids, and catecholamines. Insulin resistance is further associated with metabolic abnormalities such as diabetes, lipid dysfunction, nonalcoholic fatty liver disease (NAFLD), cardiovascular disease, retinopathy, and polycystic ovary syndrome (PCOS). Each of these factors is also interconnected, thus resulting in a potentially vicious cycle.

The presence of insulin resistance during spaceflight has been reported, and sex differences play a role, with female astronauts displaying better Homeostatic Model Assessment for Insulin Resistance (HOMA-IR) values than male astronauts[6,7]. Sex differences play a role in diseases and metabolism both on Earth and in space[8]. Some of these differences may be attributed to sex hormones. Estrogens play a variety of roles and serve many functions in women's lives. Besides being a reproductive hormone, estrogen is involved in metabolism as well as in the cardiovascular, nervous, and immune systems[9]. Estrogen signaling activates membrane and nuclear receptors in both canonical and non-canonical ways to modulate metabolism, and promote growth, proliferation, and differentiation of diverse cell types[10,11]. Estrogen receptors are ESR1, estrogen receptor beta (ESR2), and G-protein coupled estrogen receptor 1 (GPER1). There are also estrogen-related receptors alpha, beta and gamma (ESRRA, ESRRB, ESRRG) which are structurally similar to ESR1 and ESR2[12] but whose ligands are unknown. ESRRA also has DNA sequence homology to the estrogen receptor and can be activated by phytoestrogens (i.e. genistein)[13]. These receptors play pivotal roles in metabolism, insulin resistance, NAFLD, and mitochondrial function[12]. Besides that, an unbalanced ratio of these receptors may have an impact on metabolism[14]. Additionally, estrogen and insulin combine in controlling metabolic pathways. In animal models, knockout of the estrogen receptor alpha (ERα) (ESR1) in both male and female mice results in insulin resistance and dysglycemia[15–17]. Estrogen receptors promote pancreatic cell survival, enhance insulin production, and improve insulin sensitivity[18,19]. Female and male ESR1−/− mice both display reduced GLUT4 expression, further emphasizing the role of estrogen in glycemic control[14]. Clinically, compared to premenopausal women, postmenopausal women have an increased risk of adverse metabolic profiles such as type 2 diabetes, insulin resistance, impaired glucose and lipid profiles, increased android fat deposition, sarcopenia, and osteoporosis[20]. Estrogens also

positively correlate with bone and muscle mass and are protective against disuse atrophy[21]. Thus, there is substantial reason to believe that insulin and estrogen signaling are key pathways regulating health in space.

The amount of knowledge in the literature about how insulin and estrogen signaling and related pathways change during spaceflight is currently sparse. Our preliminary findings reported possible insulin and estrogen signaling alteration during spaceflight[22,23]. A few published studies show insulin resistance parameters may change during spaceflight but data are conflicting, suggesting either decreased insulin or no change[24,25]. Similarly, data on estrogen signaling are conflicting with one study revealing downregulation of ESR1 and ESR2 in the uterus[26] whereas a recent study found no alterations in estrogen receptor expression in rodent ovaries[27]. To address these gaps in our knowledge of insulin and estrogen signaling in response to spaceflight we examined these signaling pathways across multiple tissues in rodents from the same mission as well as in a group of astronauts and a set of commercial spaceflight participants/astronauts. Our results indicate significant changes in these pathways in response to spaceflight in both mice and people. These observations lay the foundation for further studies aimed at understanding the metabolic alterations induced by spaceflight as well as highlighting insulin and estrogen signaling as potential targets for countermeasures in future spaceflight missions.

## Results
**Spaceflight demonstrates global impact of insulin and estrogen signals across different organs in female mice.** To understand how insulin and estrogen signaling change during spaceflight we examined gene expression in mice that were flown to the International Space Station (ISS) for 37 days. For this purpose, adrenal glands (OSD-98), kidney (OSD-102), liver (OSD-168), eyes (OSD-100), and skeletal muscles (i.e. quadriceps (OSD-103), gastrocnemius (OSD-101), soleus (OSD-104), tibialis anterior (OSD-105), and extensor digitorum longus (OSD-99)) were utilized (Fig. 1a). The transcriptomic data (RNA-seq) was obtained for 16-week-old female mice from the NASA GeneLab platform[28] and were analyzed for insulin signaling, estrogen signaling and insulin resistance related genes and pathways. The gene lists for these hormones were obtained from the NCBI resource[29] (i.e. 'insulin signaling' mus musculus). The greatest number of differentially expressed genes (DEGs) were observed in the mouse liver ($n = 3670$), followed by soleus ($n = 3518$) and extensor digitorum longus (EDL) ($n = 1859$) muscles. In all, 2677 DEGs were unique to the liver, 2048 DEGs were unique to soleus muscle, and 784 DEGs were unique to EDL (Fig. 1b). The least number of DEGs was noted in the adrenal gland ($n = 61$) and 28 of them were unique to it (Fig. 1b). There are a total of 1970 unique genes from the NCBI curated gene list which are commonly encountered across insulin resistance, insulin signaling and estrogen signaling pathways. Most of the genes from the NCBI list related to insulin resistance, insulin signaling, and estrogen signaling were downregulated in the liver (Fig. 1c). In total, 645 out of the 1970 unique genes from the NCBI resource had gene expression values across all nine tissues. These genes were divided into seven biologically relevant clusters associated with insulin resistance and signaling (Fig. 1d). In particular, the last cluster (cluster 7) at the bottom of the heatmap were related to Gene Ontology (GO) processes[30] such as lipid localization and glycerolipid metabolic processes which are related to obesity which in turn can lead to a decrease in insulin sensitivity[31] (Supplementary Fig. 1).

In general, two genes were differentially expressed (i.e., upregulated) across all tissues in the RR-1 samples; ARNTL and

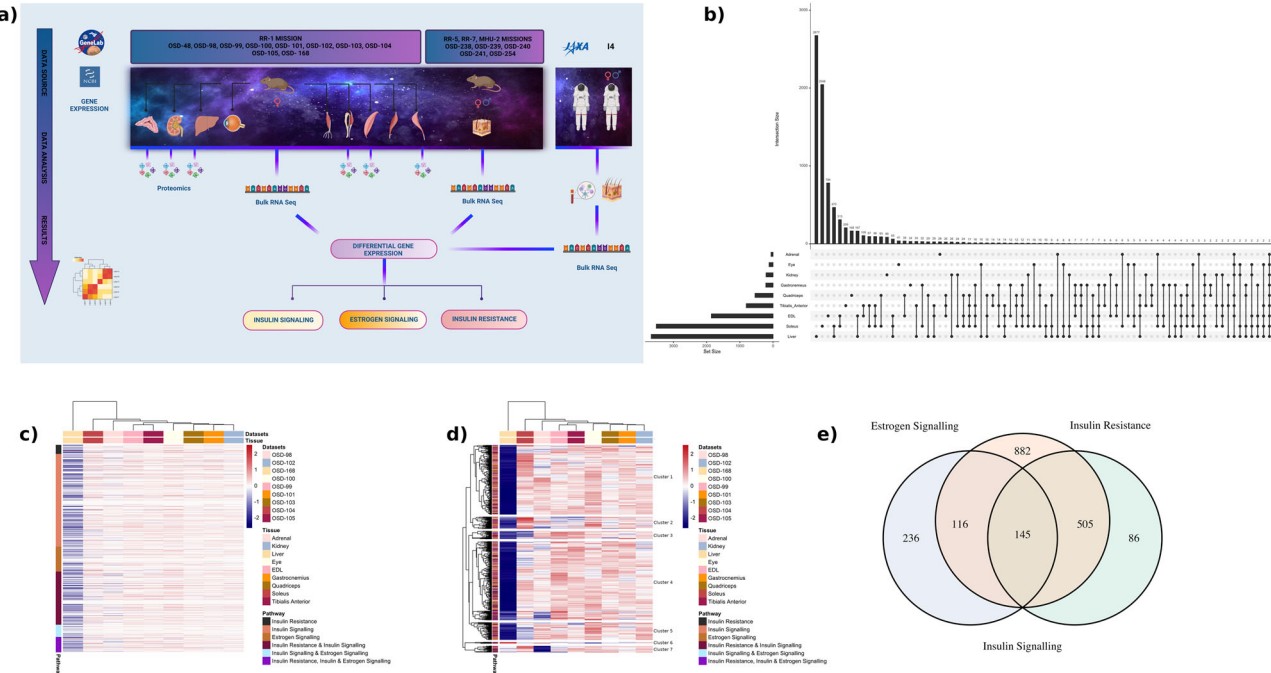

**Fig. 1 Unique differentially expressed genes across tissues in mice. a** Graphical abstract showing the samples and analysis created in BioRender. **b** UpSet plot showing the number of differentially expressed genes (FC cutoff of 1.2 and *p*-adj<0.05) that are unique to each tissue and overlapping different tissues. The liver had the highest number of unique genes while adrenal had the lowest. The two genes that were common across all nine tissues were related to circadian rhythm and insulin signaling. **c** Global gene level heatmap of 1970 unique genes from NCBI curated gene list which are common across insulin resistance, insulin signaling and estrogen signaling pathways. Genes are the rows and columns are the nine tissues analyzed from the RR1 mission. Genes were further divided into being unique to only one of these pathways or a combination of the pathways. Heatmap shows the *z*-scores of logFC values from the nine tissues and the tissues are clustered using hierarchical clustering. Liver was the most affected with genes being downregulated across all pathways when comparing flight vs pre-flight. **d** Heatmap of 645 unique genes from NCBI curated gene list which have log fold change values across all 9 tissues. Hierarchical clustering of genes shows clusters being related to GO biological processes relevant to insulin resistance, insulin signaling and estrogen signaling (Fig. S1). **e** Venn diagram of insulin resistance, insulin signaling, estrogen signaling pathways with their corresponding unique and overlapping genes from the NCBI curated gene list. There are a total of 145 genes that are common across all three pathways.

*NPAS2*. Across skeletal muscles, 21 DEGs were commonly encountered, out of which 11 DEGs were upregulated and 9 DEGs were downregulated. *UCKL1OS* was only upregulated in soleus. Many of them were related to the circadian clock, such as *NR1D2, NPAS2, PER2, PER3, CIART, DBP, ARNTL,* and *SOX4*. In terms of insulin signaling linked genes, in muscles, *ARNTL, MT1, SOX4* were upregulated and *DBP* and *PER2* were downregulated. The genes associated with circadian rhythm, indicates the global effect of spaceflight on the circadian clock and its association with insulin signaling. Overall, from our analysis, the liver was the most impacted organ by spaceflight, with the highest number of DEGs, alongside insulin and estrogen receptor signaling predicted to be inhibited (*z*-scores: −3.772 and −5.859, respectively).

**Common insulin and estrogen signaling genes altered by spaceflight in female rodents.** To determine if there are any universally shared insulin and estrogen pathways being regulated throughout the different organs during spaceflight, we performed further analysis on the 145 shared genes linked to insulin and estrogen signaling as well as insulin resistance (Fig. 1e). 56 out of 145 common genes across the insulin resistance and insulin signaling pathways had gene expression values across all nine tissues. The genes were clustered into three clusters where the top enriched pathways from GO processes were: triglyceride-rich lipoprotein particle remodeling for cluster 1, positive regulation of insulin secretion for cluster 2, and carbohydrate homeostasis for cluster 3 (Fig. 2a). The disease enrichment analysis revealed

that 62 out of 145 genes were associated with diseases, with the top enriched disease being diabetes mellitus, followed by fatty liver (Fig. 2b). Human genes at the intersection of these pathways were used in a similar manner, and the top enriched disease was obesity, followed by hypertension (Fig. 2c). Thus, enriched diseases in both species emphasize the importance of insulin and estrogen signaling in metabolic diseases.

Further pathway analysis utilizing gene set enrichment analysis (GSEA), focused on specific insulin and estrogen pathways, demonstrated differences in the enrichment of gene sets across organs (Fig. 3a). The majority of gene sets related to insulin and insulin resistance were predicted to be inhibited in the liver (OSD-168) such as insulin receptor signaling pathway, response to troglitazone and adipogenesis (Fig. 3a). Kidney (OSD-102) and eye (OSD-100) also showed similar inhibition for insulin pathways, such as insulin pathway, insulin receptor signaling pathway, response to insulin (Fig. 3A). Interestingly, a set of muscle tissues (i.e. tibialis anterior (OSD-105) and EDL (OSD-99)) were shown to be activated for response to insulin, while the quadriceps (OSD-103) and gastrocnemius (OSD-101) showed negative enrichment scores for response to insulin (Fig. 3a). It is known that during spaceflight the soleus muscle is impacted first with muscle mass loss which can lead to downstream dysregulation related to metabolic, inflammatory, and immune functions[32].

Overall, there are similar organ dependent patterns for estrogen-specific pathways observed (Fig. 3a). Estrogen metabolism and signaling pathways were negatively enriched in the kidney (OSD-102), liver (OSD-168), and quadriceps muscle

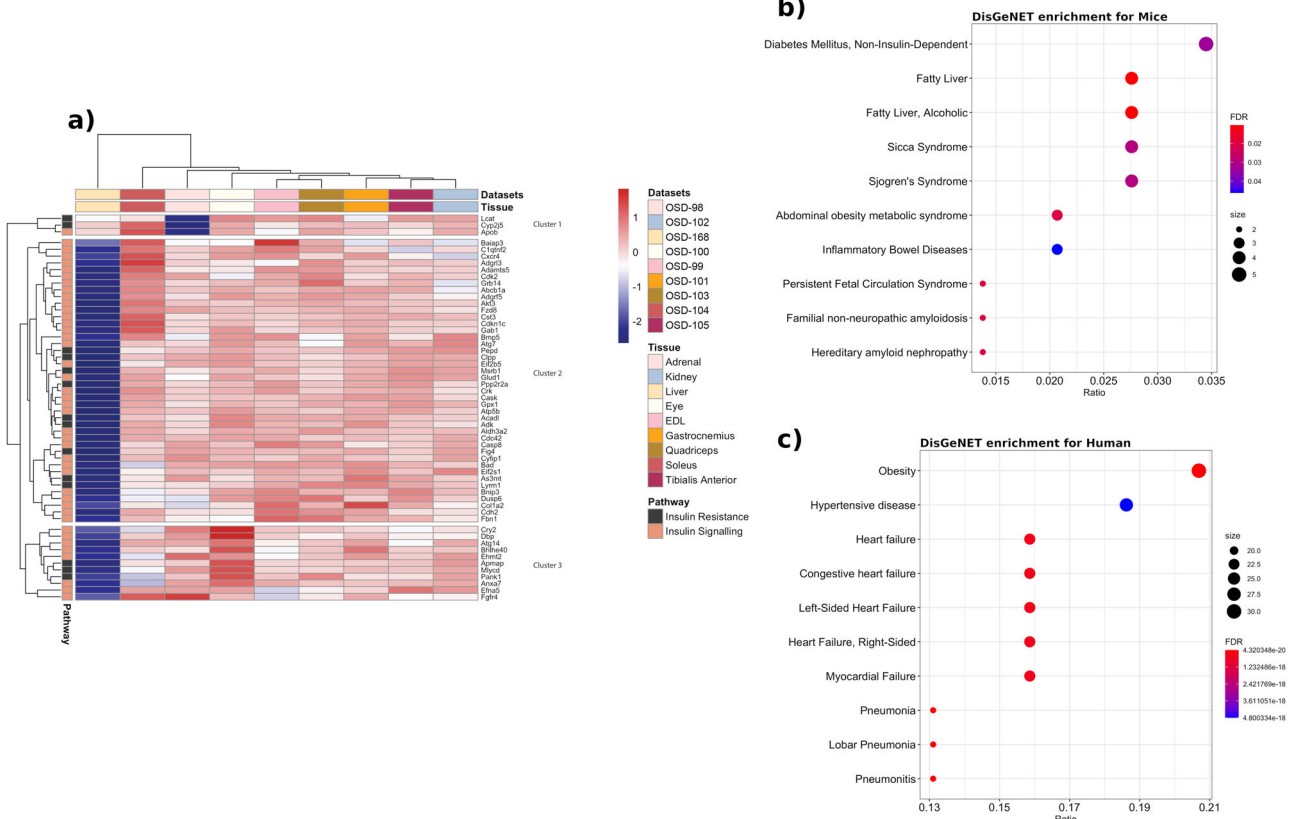

**Fig. 2 Insulin resistance, insulin and estrogen signaling related genes and diseases. a** Heatmap for 56 out of 145 common genes across the insulin resistance and insulin signaling pathways for which there were gene expression values across all nine tissues. **b** Disease enrichment plot showing top 10 diseases from mouse genome database (MGD) being enriched using 145 unique genes common across insulin resistance, insulin signaling and estrogen signaling. The y axis shows the disease labels and the x axis shows the ratio of disease relevant genes to the total number of genes found from the 145 genes that were present in MGD. The dots in the enrichment plots are colored by the FDR values and the sizes of the dots indicate the number of genes found to be enriched for a disease. Diabetes and fatty liver are the top two enriched diseases which are related to estrogen and insulin signaling. **c** Disease enrichment plot is shown with top 10 diseases from human curated sources (e.g., UniProt, ClinGen) using species relevant genes out of 145 genes from the NCBI list. Obesity and fatty liver are the top two enriched diseases which are also related to estrogen and insulin signaling.

(OSD-103). Specific estrogen signaling pathways, such as pubertal breast, were negatively enriched in the liver, eye, and adrenal gland, while being positively enriched in the (OSD-99) and tibialis anterior (OSD-105) muscles (Fig. 3a).

When analyzing the specific leading-edge genes involved in the insulin and estrogen pathway predictions, we observed that a subset of these genes were related to the circadian rhythm (i.e., *ARNTL, CLOCK, CRY2, PER3, NFIL3*, and *NR1D2*) (Fig. 3b). The majority of these gene expressions were in the same direction across organs; however, some unique differences were observed. *PFKM* was upregulated only in the soleus muscle (OSD-104) (Fig. 3b).

**Gene expression changes and proteomics suggest underlying insulin resistance in spaceflight-induced hepatosteatosis.** Notably, the liver is a key metabolic organ that orchestrates and regulates a plethora of metabolic pathways, including estrogen and insulin signaling[33]. During spaceflight, the liver RNA-seq data exhibits the most pronounced and unique impact on metabolic pathways as well as insulin and estrogen signaling. A comprehensive analysis of liver samples harvested after 37 days of spaceflight, revealed that both insulin and estrogen receptor signaling are predicted to be inhibited in the liver during spaceflight, as compared to ground controls (z-scores: −3.772 and −5.859, respectively). Our findings indicate that several genes of the insulin signaling pathway denoted development of insulin

resistance and hepatic steatosis (z-scores: 2.481 and 3.737, respectively). Furthermore, insulin receptor was downregulated in the liver during spaceflight, along with inhibition of mitochondrial fatty acid oxidation (FAO), which may indicate derangement of mitochondrial activity. In addition, disruption of estrogen signaling has been extensively reported to aggravate hepatic insulin resistance[33,34]. According to this evidence, our analysis showed inhibition of estrogen receptor signaling in the liver during spaceflight (z-score: −5.859). Remarkably, genes associated with mitochondrial complex I, complex II, and ATP synthase were downregulated in the estrogen signaling network (Fig. 4a). This is particularly relevant considering that estrogen receptors play a key role in mitochondrial activity and biogenesis[35–37]. In livers harvested from mice after spaceflight, we observed impairment of estrogen signaling, which in turn may hamper oxidative phosphorylation by mainly affecting complex I (Fig. 4a)[38].

Specific analysis of pathways related to hepatic steatosis (Fig. 4b, c) revealed that the SREBP pathway was one of the top positively enriched pathways (Fig. 4b). This pathway is known to be involved in metabolic-associated chronic liver diseases such as Nonalcoholic Fatty Liver Disease (NAFLD)[39]. Furthermore, response to insulin and insulin signaling pathways were universally enriched in hepatic steatosis during spaceflight, with insulin receptor (*INSR*) and downstream targets (i.e *IRS1, AKT1, AKT2*) among the top enriched genes for hepatic steatosis

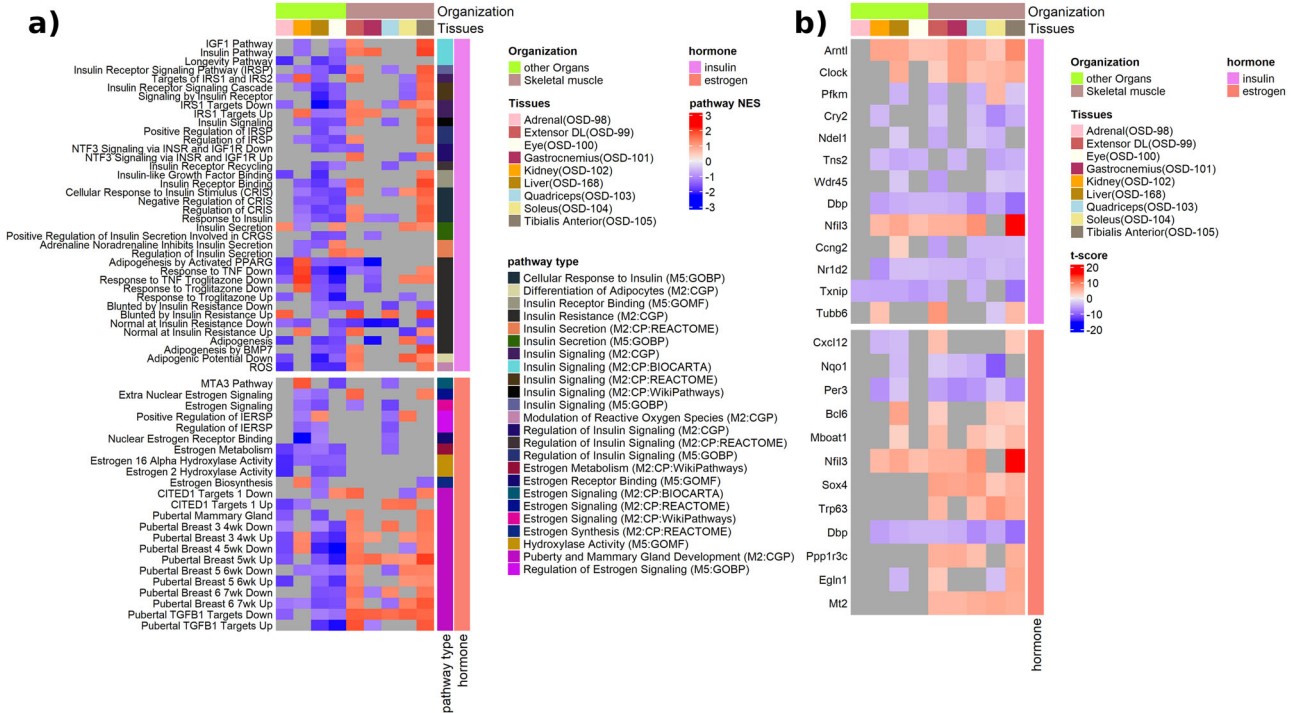

**Fig. 3 Gene set enrichment analysis of insulin and estrogen pathways. a** A heatmap of the normalized enrichment scores (NES) of estrogen and insulin pathways impacted by spaceflight in datasets of RR-1 mission samples. The dark gray locations in the heatmap indicate missing values for the NES, resulting from off-range adjusted p-values (padj) at which the NES is obtained. The assumed range is padj < 0.3. A positive NES indicates activation of the pathways, while at negative NES the pathway is inhibited. In the pathways, the full meaning of CRGS is 'Cellular Response to Glucose Stimulus'; and the full meaning of IERSP is 'Intracellular Estrogen Receptor Signaling Pathway'. The pathways are categorized into several types, and the MSigDB database from which the pathway is sourced is indicated in parentheses. **b** A heatmap of the t-score of the overlapping leading-edge pathway genes across the datasets of the RR-1 mission samples. The dark gray locations in the heatmap indicate the absence of a value for the t-score, resulting from off-range adjusted p-values (padj) at which the t-score is obtained. The assumed range is padj < 0.05. The leading-edge genes are the genes that contribute to the GSEA predicted pathways. The overlapping leading-edge genes are commonly leading-edge genes to at least four of the datasets.

(Fig. 4c). Among the top leading genes, we additionally identified *MFN2*, which encodes for Mitofusin 2, a mitochondrial fusion protein associated with insulin signaling and oxidative stress in NAFLD[40,41]. In summary, our findings suggest that during spaceflight, alterations of insulin and estrogen signaling may lead to the development of hepatic steatosis.

We also performed proteomics analysis using RR-1 data, which revealed that the liver displayed the most insulin and estrogen-related protein abundance changes (Fig. 5a). In the liver, proteins involved in lipid metabolism such as hormone sensitive lipase, Apoe, Sphk2 and Ldlr, showed notable changes in protein abundance. These proteins are also linked to insulin signaling, insulin resistance, hepatic steatosis and glucose metabolism disorder (Fig. 5b–e). *Lipe* encodes hormone sensitive lipase (HSL) which hydrolyses lipids (i.e. triacylglycerols) in various tissues. Insulin inhibits HSL activity while counteracting hormones such as glucocorticoids, glucagon, and catecholamines activate HSL during stress or starvation[42]. Increased ApoE protein abundance was observed in the kidney, adrenal gland, and significantly in the liver (Fig. 5c, d). Apolipoprotein E (ApoE) is one of the major lipoproteins in the cholesterol metabolism and it uptakes VLDL, IDL and chylomicron remnants from the circulation. It is also a major ligand for the LDL receptor. There was a significant increase in the abundance of LDL receptors in the liver, while it decreased in the adrenal gland (Fig. 5c–e). In terms of estrogen signaling related proteins, we observed changes in protein abundance associated with mitochondrial complexes (Fig. 5f). This was observed mostly in muscles, and the majority of these mitochondrial proteins belong to NADH dehydrogenases

in complex I (i.e. *NDUFA5* encodes a subunit of complex I) (Fig. 5f).

**Alterations in insulin and estrogen linked genes in the rodent skin.** Both insulin and estrogen act in the skin and contribute to important functions such as wound healing and skin integrity. Further, certain skin conditions have been associated with insulin resistance (i.e., acanthosis nigricans)[43]. Thus, we utilized rodent data sets (RR-5, RR-7, MHU-2) to analyze skin gene expressions linked to insulin resistance, insulin, and estrogen signaling. We observed differences in pathway enrichment scores between female and male mice (Fig. 6a). During the first 25 days of flight, estrogen and insulin signaling, as well as insulin resistance pathways, were negatively enriched in the skin of female C3H mice; however, these pathways were positively enriched during the 75 days of flight (Fig. 6a). We also noted similarities and differences across strains. Within the same mission (RR-7), pathway enrichments in C3H mice varied based on flight duration, while the insulin resistance pathway was positively enriched in C57BL mice regardless of flight duration (Fig. 6a). The insulin resistance pathway was positively enriched during 75 days of flight in two different strains (Fig. 6a). Further, in female mice, the insulin resistance pathway was positively enriched in various strains and different missions, except in C3H strains exposed to 25 days of flight (Fig. 6a). In the skin of male mice, differences in enrichment scores were observed based on the tissue (dorsal vs femoral skin) and diet (Fig. 6a).

In female and male rodent skin samples, we found differences in gene expressions related to insulin and estrogen signaling

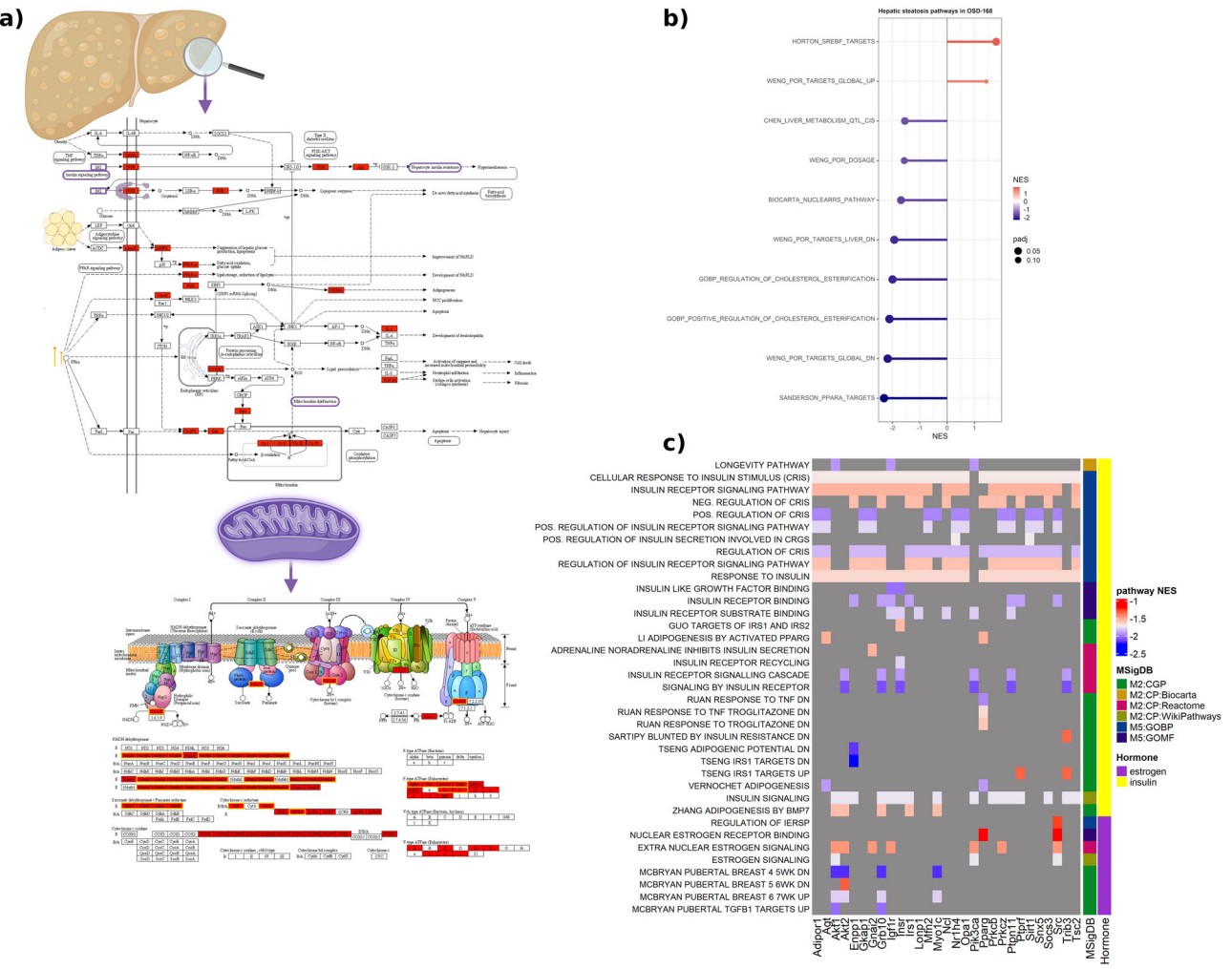

**Fig. 4 Hepatic steatosis in mice liver. a** The hepatic steatosis canonical pathway with the significantly regulated genes for liver tissue from mice flown to space compared to ground control mice. The lower panel is the mitochondrial complexes and the significantly regulated genes involved visualized by ShinyGo v0.76.3 and KEGG pathway diagrams. The boxes highlighted in yellow indicate genes related to estrogen receptor signaling. Panel was created in BioRender. **b** Lollipop plot illustrating the normalized enrichment scores (NES) of hepatic steatosis pathways in OSD-168 (liver tissue). Only pathways with a NES obtained at an adjusted *p*-value <0.3 (padj < 0.3) are shown. **c** An illustration of common leading-edge genes across pathways of insulin and estrogen in a heatmap of the normalized enrichment scores (NES) of the pathways impacted by spaceflight in OSD-168 (liver tissue). Only pathways with NES at padj < 0.3 are shown, and only genes that are among leading edge genes seven (7) or more of the pathways are shown.

(Fig. 6b). Most of the significant up- and downregulations were observed in the male mice, especially in the dorsal skin (Fig. 6B). Certain genes involved in tissue remodeling, extracellular matrix, wound healing, and collagen synthesis (i.e., *ACE, FN1, IGFBP5, MMP3, COL6A3, COL6A1, COL1A1, COL5A1, BGN, FASN, LOXL2*) were downregulated only in male mice (Fig. 6b). *FN1* encodes fibronectin which is an essential component of the extracellular matrix (ECM) and participates in cell migration, adhesion, and wound repair[44]. Matrix metalloproteinases (MMPs) are involved in tissue remodeling; MMP3 regulates the rate of wound healing and was downregulated in the skin of male mice[45]. Further, epidermal lipids contribute to the skin's barrier function[46]. Fatty acids can be synthesized in the epidermis, and *FASN* encodes fatty acid synthase, which was downregulated in male mice. Insulin receptor (*INSR*), *IRS2*, estrogen receptor 1 (*ESR1*), and *ESRRG* were upregulated in male mice dorsal skin (Fig. 6b). Estrogen promotes skin and mucosa epithelialization[47,48]. Sex differences may play a role in wound healing and variations in healing rates have been reported in literature[49,50]. In female mice, *S100s9, Nr4a2, Nr4a3, Il6*, and *Egf2* were significantly downregulated only in the skin of C3H mice

during 25 days of flight (Fig. 6b). Although we noticed some changes in the pathway enrichment scores, results need to be interpreted carefully, due to differences in missions, durations, strains, housing, and diet.

**Changes in insulin signaling genes and pathways conserved in astronauts.** To explore if insulin and estrogen signaling-related genes are affected in astronauts during spaceflight, human samples were used to map genes of interest. We utilized astronaut data from the Japanese Space Agency (JAXA) Cell-Free Epigenome (CFE) Study mission[51] with astronauts in space for 120 days. Specifically, for the JAXA mission RNA-seq was conducted on plasma cell-free RNAs (cfRNAs) from 6 astronauts that were on the International Space Station (ISS) for 120 days from pre-flight, during flight, and post flight time points.

Insulin signaling linked genes are dysregulated during and after spaceflight in astronauts (Fig. 7a). Some genes were specifically upregulated within the first month upon return (R + 3d, R + 30d) such as *AVP, FFAR4, GLP1R, RIMS2, FGF6, NR1H4, PRLH, CPE*, and *APOM*. The *FFAR4* encodes free fatty acid receptor 4, is widely expressed in tissues and has been associated with

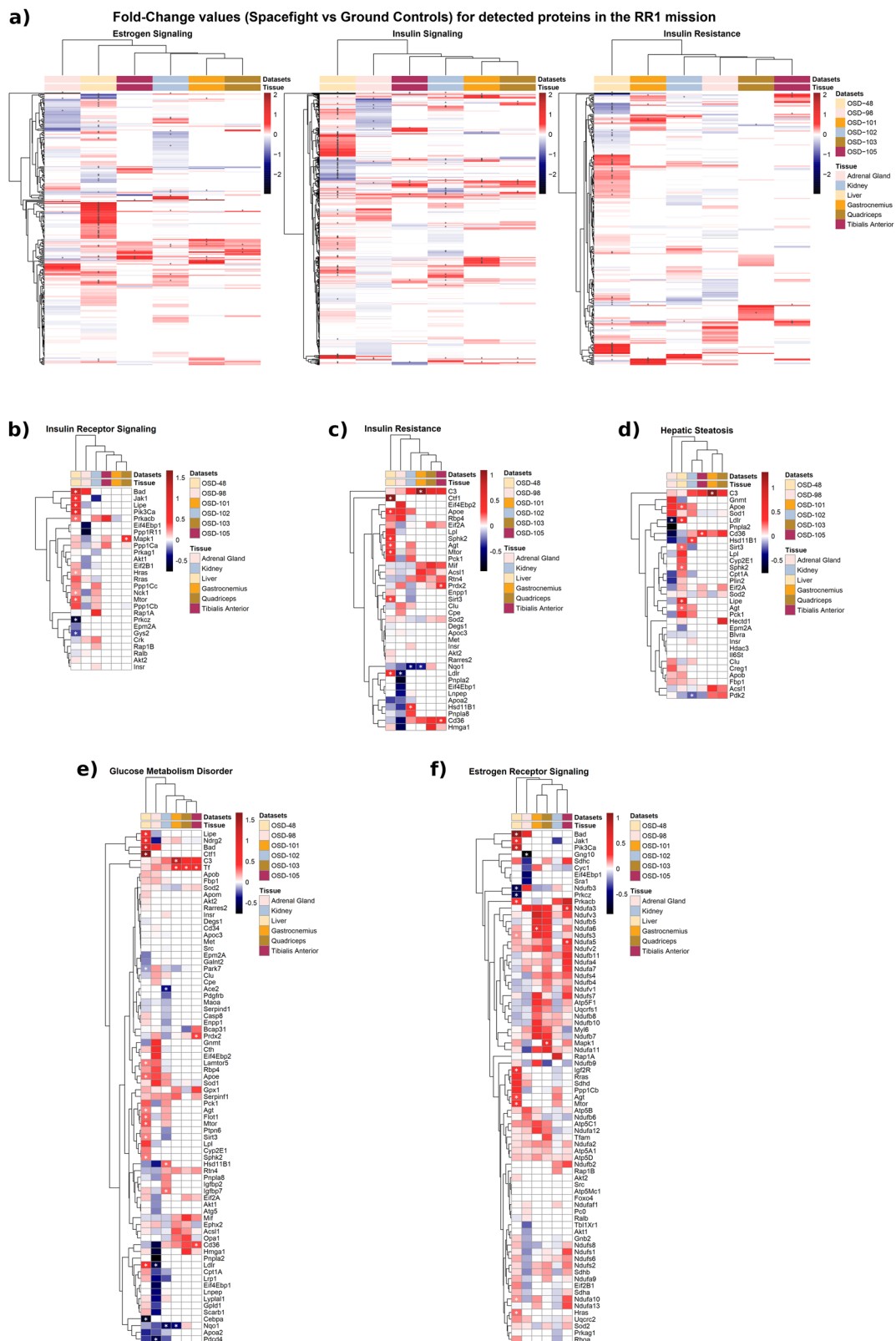

**Fig. 5 Quantitative proteomics screening of different tissues from the RR1 mission samples. a** Heatmaps for logFC (Flight vs. Ground) of proteins across the estrogen signaling, insulin signaling, and insulin resistance pathways. Normalized protein intensity values were used to calculate the logFC through the Limma package. Columns represent datasets from different tissues in the RR1 missions, and rows correspond to proteins. The gray cells in the heatmap indicate missing values. Asterisks (*) overlaying cells denote statistical significance (p-value < 0.05). Liver displayed the most protein abundance changes across these pathways. Protein abundance changes for (**b**) insulin receptor signaling, (**c**) insulin resistance, (**d**) hepatic steatosis, (**e**) glucose metabolism disorders, and (**f**) estrogen receptor signaling, were also depicted in different heatmaps. All 0 values for **b**–**f** are for proteins which were not present in the data.

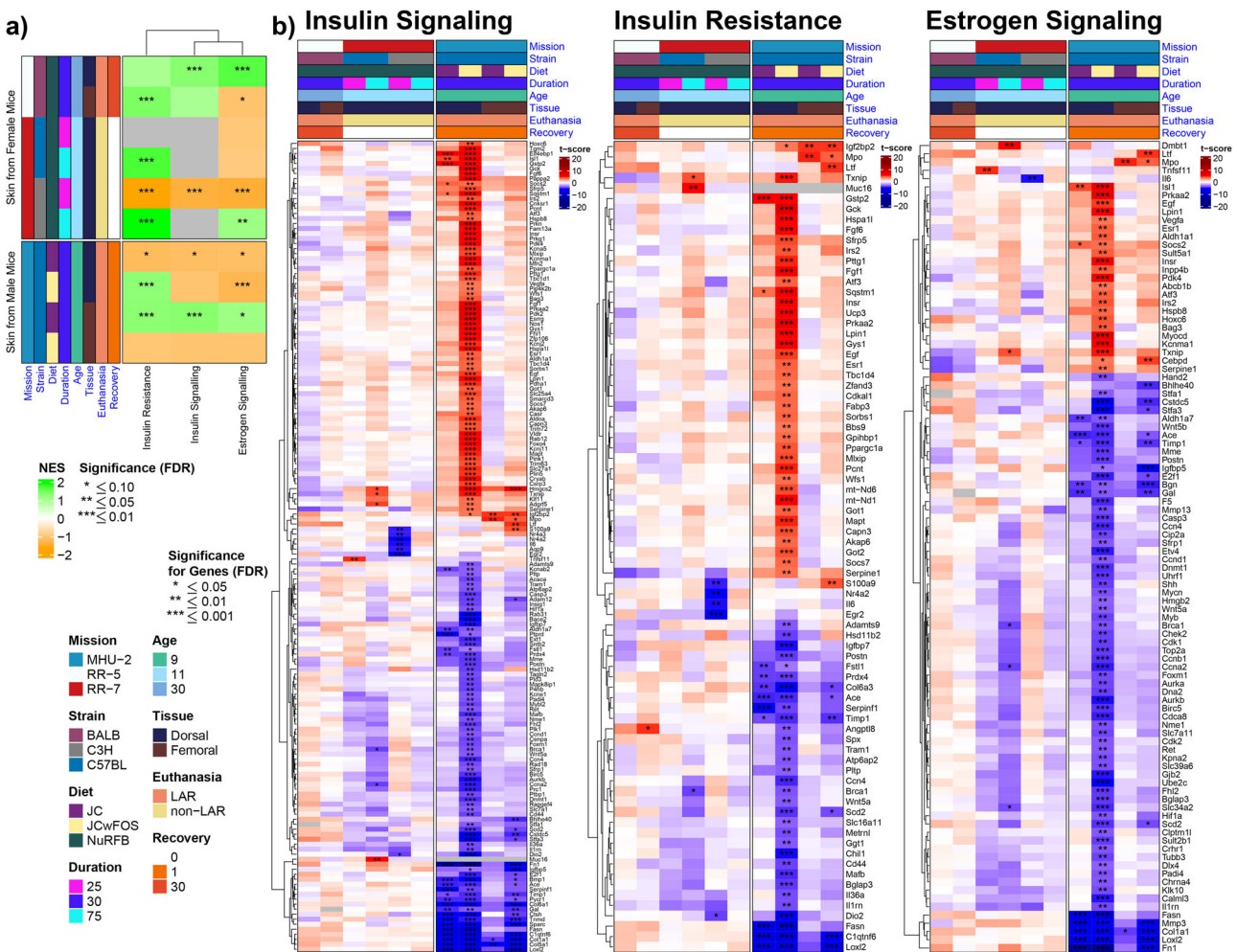

**Fig. 6 Insulin and estrogen signaling, insulin resistance pathways in skin rodent samples. a** Pathway enrichment heatmap. Heatmap of normalized enrichment scores for insulin resistance, insulin signaling, and estrogen signaling pathways in spaceflight vs ground control comparisons of murine skin. Columns represent pathways, while rows represent datasets from multiple spaceflight missions, split by biological sex. Asterisks (*) overlaying cells denote statistical significance. **b** Gene-level heatmaps (one for each pathway). Heatmap of t-scores for highly significant (FDR ≤ 0.01) differentially expressed genes associated with the insulin and estrogen signaling as well as insulin resistance. Columns represent datasets from multiple spaceflight missions, and rows correspond to genes. Asterisks (*) overlaying cells denote statistical significance; genes must be significant in at least one of the datasets to be displayed.

metabolic and cardiovascular diseases[52]. The *GLP1R* is another key gene encoding glucagon-like peptide (GLP-1) receptor which stimulates insulin secretion (Fig. 7a). Additionally, insulin secretion was among the enriched pathways (Fig. 7a). GLP-1 receptor agonist is currently used in the treatment of diabetes and obesity[53].

Certain genes involved in insulin signaling were upregulated throughout the postflight period. *MAF* is a transcription factor that is known to regulate and activate insulin gene expression[54] (Fig. 7a). Similarly, *RARRES2*, which encodes chemerin, was also upregulated throughout postflight. Chemerin is an adipokine that is associated with insulin resistance and obesity[55]. *AKT1* is the main downstream of insulin signaling and was upregulated during postflight which also correlates with the postflight PI3K/AKT enrichment (Fig. 7a). *CRH, GCGR*, and *SLC2A3* are other important genes related to endocrine hormones and insulin. The glucagon receptor, encoded by *GCGR*, is the primary receptor for glucagon. Glucagon counteracts the actions of insulin and has a role in stress response[56]. The *GCGR* was upregulated throughout the entire postflight period (Fig. 7b). *SLC2A3* encodes for a glucose transporter named GLUT3 and is widely expressed in various cells. *SLC2A3* was downregulated upon return from

spaceflight (Fig. 7b). The cholesterol metabolism was the top activated pathway for postflight upregulated genes involved in insulin signaling (Fig. 7a). Insulin secretion, regulation of lipolysis in adipocytes, Cushing syndrome, and circadian entrainment were other enriched pathways (Fig. 7a).

Glucagon signaling was the top activated pathway for down-regulated genes associated with insulin signaling (Fig. 7b). Insulin resistance, lipid and atherosclerosis, AMPK signaling, oxytocin signaling and viral infection related pathways (i.e., HPV, CMV infections, HTLV1) were other enriched pathways (Fig. 7b). The *MS4A1* gene encodes CD20, a B-cell marker, and it was downregulated during postflight but upregulated during flight (Fig. 7b).

Further, some of the gene expressions changed at specific time points. Interestingly, *CRP* was upregulated upon return (R + 3d) and was downregulated a month later (R + 30d) (Fig. 7a). Follow-up samples showed upregulation of *CRP* on the second and third months postflight (Fig. 7a). *CRP* is also linked to estrogen signaling and insulin resistance (Figs. 8a, 9a). This is important as the *CRP* gene encodes c-reactive protein, which is an inflammatory marker. Elevated CRP levels correlate with both fasting insulin and insulin resistance parameters (i.e. HOMA-IR)[57].

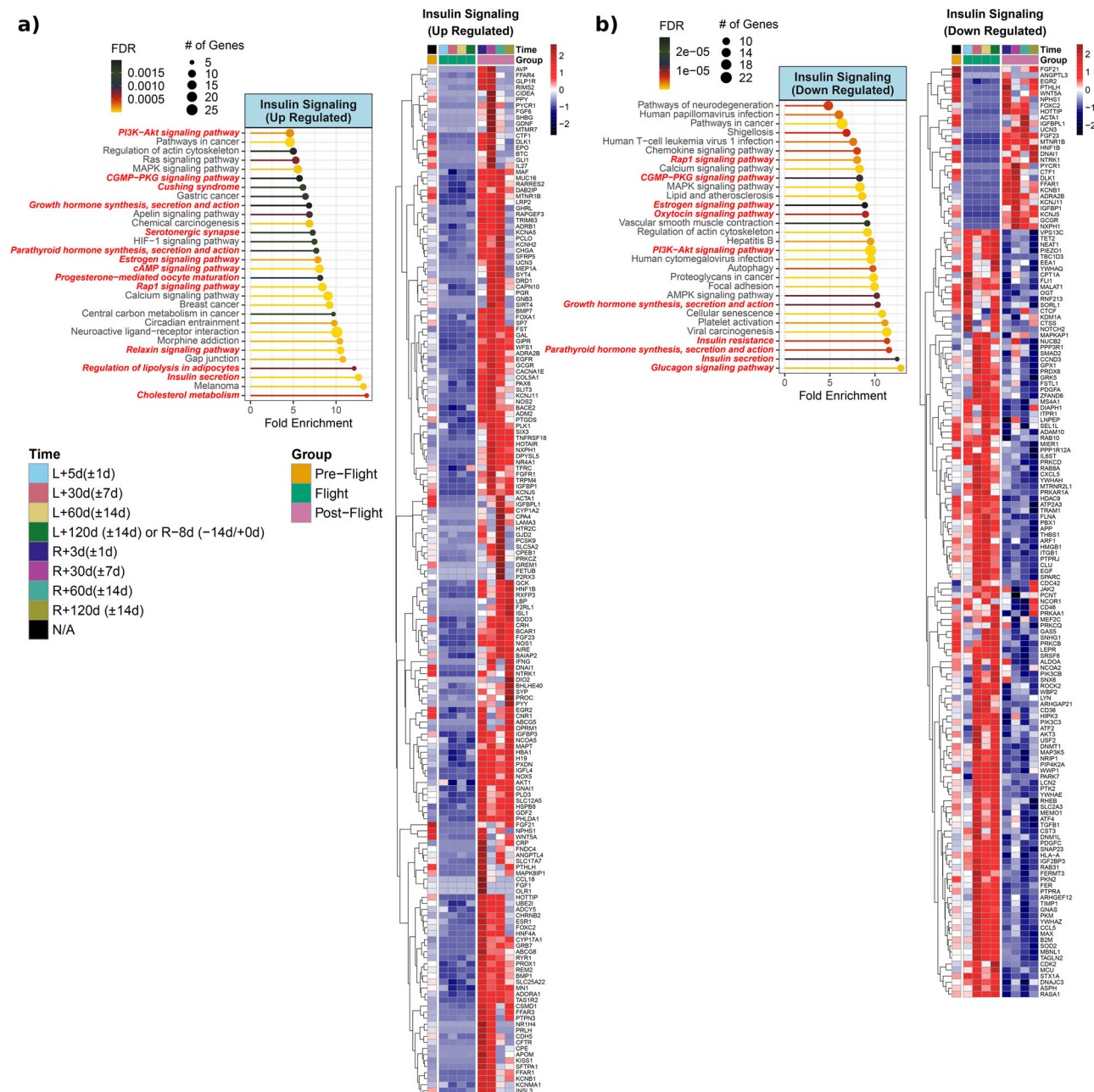

**Fig. 7 Insulin Signaling linked genes and pathways from JAXA Cell-Free Epigenome (CFE). a** Heatmap of the significantly upregulated genes compared to pre-flight for either flight or post-flight of normalized plasma cell-free RNA expression values and associated pathways for insulin signaling over time for the six astronauts over 120 days in space from JAXA study. The values shown on are the averaged normalized expression values for all six astronauts each time point during flight and post-flight. The four pre-flight time points were averaged together, since the changes for genes in the time leading up to flight are considered to be the same and part of the baseline values. For the time, L = Launch (i.e., meaning time after launch from Earth and length in space) and R = Return to Earth. Based on heatmap, pathway analysis performed using ShinyGO to reveal the top 30 pathways (FDR < 0.05) being regulated for postflight upregulated insulin linked genes. The red bold font texts are the pathways that are directly known to regulate the insulin pathway in the pathway figure. **b** Heatmap and associated pathways showing insulin signaling linked downregulated genes in postflight or flight vs preflight. Based on heatmap, pathway analysis performed using ShinyGO to reveal which pathways are being regulated for downregulated insulin linked genes. The pathways and lollipop plots demonstrate top 30 pathways (based on FDR). The red bold font texts are the pathways that are directly known to regulate the insulin pathway in the pathway figure.

**Insulin Resistance related genes and pathways during and after spaceflight.** The majority of insulin signaling related genes and their alterations could result in insulin resistance which would be a great concern during long-term spaceflight. *AKT1* was upregulated and *AKT3* was downregulated throughout the postflight period (Fig. 8a, b). *AKT1* and *AKT3* are members of serine/threonine kinases and are downstream of insulin signaling.

Insulin resistance can be seen with hormonal dysfunctions like acromegaly, hyperaldosteronism, and Cushing disease[58]. Aldosterone is responsible for fluid and sodium balance and *CYP11B2*, and encodes aldosterone synthase, which was upregulated within the first week upon return from spaceflight (R + 3d) (Fig. 8a), a period that overlaps with intravascular fluid changes upon return. Other genes associated with insulin resistance are *MT-ND1* and

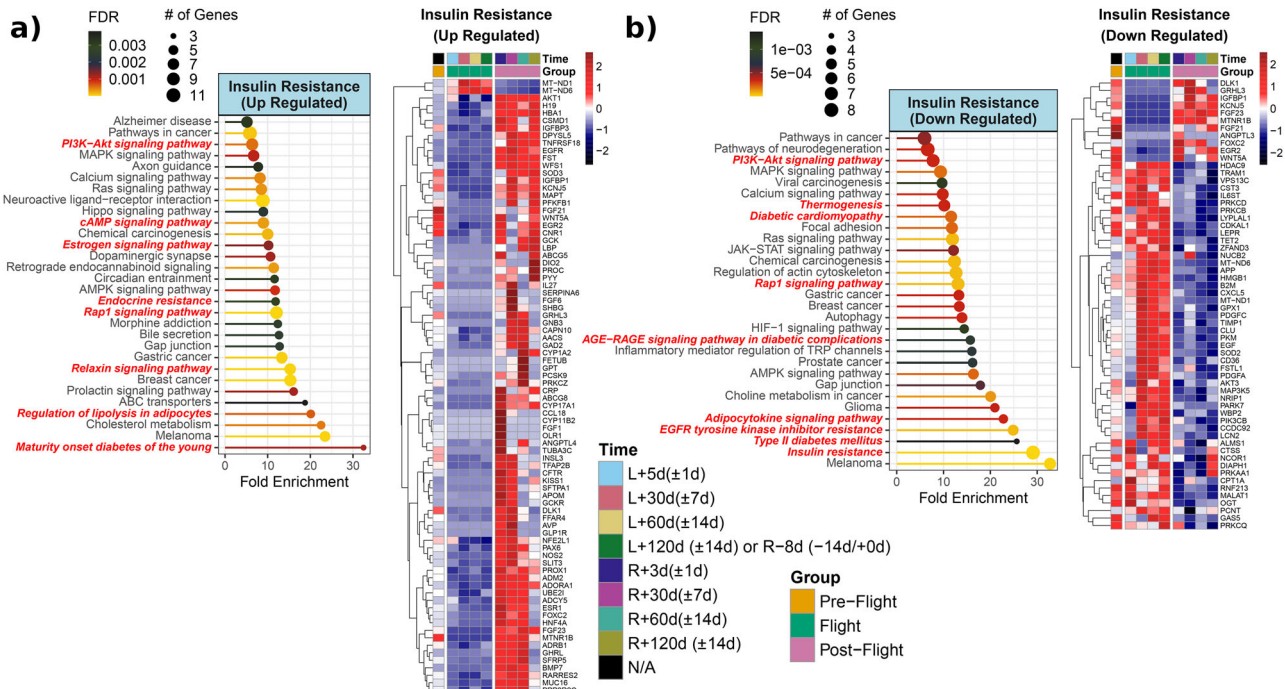

**Fig. 8 Insulin Resistant linked genes and pathways from JAXA Cell-Free Epigenome (CFE). a** Heatmap of the significantly upregulated genes compared to pre-flight for either flight or post-flight of normalized plasma cell-free RNA expression values and associated pathways for insulin resistant over time for the six astronauts over 120 days in space from JAXA study. The values shown on are the averaged normalized expression values for all six astronauts each time point during flight and post-flight. The four pre-flight time points were averaged together, since the changes for genes in the time leading up to flight are considered to be the same and part of the baseline values. For the time, L = Launch (i.e., meaning time after launch from Earth and length in space) and R = Return to Earth. Based on heatmap, pathway analysis performed using ShinyGO to reveal the top 30 pathways (FDR < 0.05) being regulated for postflight upregulated insulin-resistant linked genes. The red bold font texts are the pathways that are directly known to regulate the insulin pathway in the pathway figure. **b** Heatmap and associated pathways showing insulin resistant linked downregulated genes in postflight or flight vs preflight. Based on heatmap, pathway analysis performed using ShinyGO to reveal which pathways are being regulated for downregulated insulin-linked genes. The pathways and lollipop plots demonstrate top 30 pathways (based on FDR). The red bold font texts are the pathways that are directly known to regulate the insulin-resistant pathway in the pathway figure.

*MT-ND6. MT-ND1*, and *MT-ND6* genes encode subunits of NADH dehydrogenase in mitochondria complex I. Both genes were upregulated during spaceflight (Fig. 8a, b). Mitochondria are the center of many biochemical pathways, and one of them is fatty acid oxidation where insulin and estrogen exert their effects. *CD36* and *CPT1A* are other key genes in fatty acid metabolism and were upregulated during the various phases of spaceflight (Fig. 8b). The CD36 is responsible for fatty acid uptake into cells and is associated with insulin resistance and cardiovascular disease. *LEPR, APP, CXCL5, LYPLAL1, GHL3* are other upregulated insulin resistance related genes during flight (Fig. 8b).

In the pathway analysis, postflight upregulated genes linked to insulin resistance, showed enrichment of endocrine resistance, estrogen signaling, cholesterol metabolism, regulation of lipolysis in adipocytes, and maturity onset diabetes of the young (MODY) pathways (Fig. 8a). Downregulated genes in flight or postflight vs preflight are given in Fig. 8b. There was a trend of upregulation of these genes during spaceflight (Fig. 8b). These genes showed enriched pathways of insulin resistance, type II diabetes mellitus, AGE-RAGE signaling pathway in diabetic complications and diabetic cardiomyopathy (Fig. 8b).

The genes involved in insulin signaling and resistance from human data revolve around key cellular metabolic events such as lipid metabolism, viral infection pathways, and estrogen signaling. The endocrine-metabolic signature observed here illustrates the prominence of regulation of action of insulin during and after spaceflight. Additionally, there is a prominent immune-mediated cytokine and adipokine response to spaceflight in humans. This corroborates growth inducing actions of insulin via the prominent PI3K/Akt and the Ras/MAPK pathways.

**Estrogen signaling related genes and their pathways during and after spaceflight.** *ESR1* encodes Estrogen Receptor 1 (Estrogen Receptor alpha), the primary receptor for estrogens, which is widely expressed in both rodents and humans. Estrogen signaling via this receptor can alter energy metabolism and conversely, gonadal estrogen production is regulated by the state of energy. ESR1-regulated pathways include insulin signaling, insulin resistance, leptin signaling, and fatty acid synthesis. *ESR1* was upregulated within 2 months (R + 3d, R + 30d, R + 60d) and then downregulated again in the 3rd month (R + 120d) postflight (Fig. 9a). A similar trend during postflight was observed for *PER1* expressions (Fig. 9a). *PER1* encodes Per1 protein, which is one of the major components of circadian rhythm. Circadian rhythm and circadian clock genes are closely related to metabolism and hormones. *CYP17A1* was upregulated during postflight, which encodes a key enzyme in steroidogenesis for cortisol and sex steroid synthesis (Fig. 9a).

Estrogen-related genes altered postflight, with most being upregulated throughout the postflight period (Fig. 9a), included *ZNF423, GAL, EGFR, FST, PAX2, SLC34A1, GREB1, F5, FGF23, AKT1 and NOS1*. These genes are also implicated in the constitutive PIK3CA signaling pathway and myogenesis. This could be a direct effect of spaceflight-induced physical stress as nuclear-initiated estrogen signaling (NICE) can induce corticotropin-releasing hormone CRH through the induction of

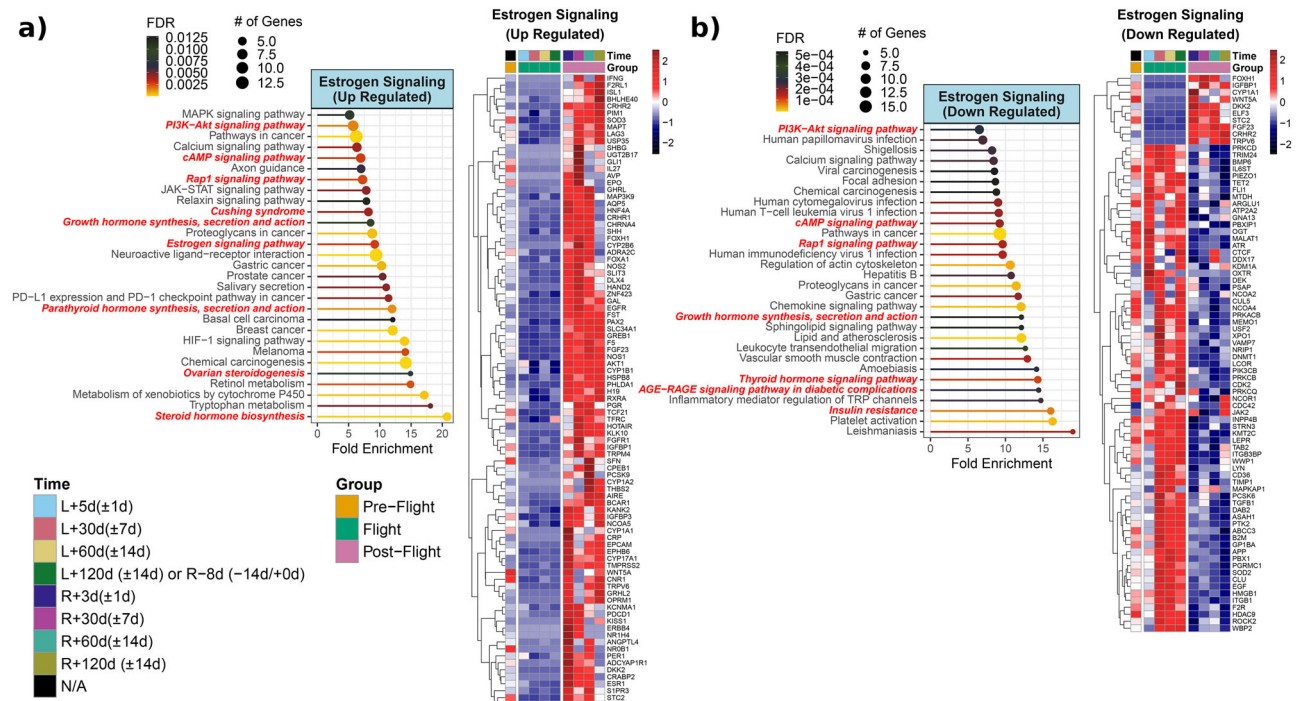

**Fig. 9 Estrogen Signaling linked genes and pathways from JAXA Cell-Free Epigenome (CFE). a** Heatmap of the significantly upregulated genes compared to pre-flight for either flight or post-flight of normalized plasma cell-free RNA expression values and associated pathways for estrogen signaling over time for the six astronauts over 120 days in space from JAXA study. The values shown on are the averaged normalized expression values for all six astronauts each time point during flight and post-flight. The four pre-flight time points were averaged together, since the changes for genes in the time leading up to flight are considered to be the same and part of the baseline values. For the time, L = Launch (i.e., meaning time after launch from Earth and length in space) and R = Return to Earth. Based on heatmap, pathway analysis performed using ShinyGO to reveal the top 30 pathways (FDR < 0.05) being regulated for postflight upregulated estrogen signaling linked genes. The red bold font texts are the pathways that are directly known to regulate the insulin pathway in the pathway figure. **b** Heatmap and associated pathways showing estrogen signaling linked downregulated genes in postflight or flight vs preflight. Based on heatmap, pathway analysis performed using ShinyGO to reveal which pathways are being regulated for downregulated estrogen signaling linked genes. The pathways and lollipop plots demonstrate top 30 pathways (based on FDR). The red bold font texts are the pathways that are directly known to regulate the estrogen signaling pathway in the pathway figure.

nitric oxide (*NOS1* upregulation) and via PI3K signaling activation[59]. Similarly, selected genes *CRHR2, PRKCD, TRIM24, IL6ST, MALAT1, OXTR, INPP4B, KMT2C, LEPR*, and *TAB2* within the downregulated cohort revealing a consistent pattern of change with spaceflight represent a niche of interleukin 6/JAK/STAT3 and MAPK signaling.

Estrogen-linked upregulated genes at postflight showed that steroid hormone biosynthesis was the top activated pathway (Fig. 9a). Other enriched pathways were ovarian steroidogenesis, estrogen signaling, Cushing syndrome, and retinol metabolism in favor of stress response and sex steroid synthesis. Estrogen-linked downregulated genes and their associated pathways showed activation of insulin resistance, AGE-RAGE signaling pathway in diabetic complications, lipid and atherosclerosis, and growth hormone synthesis pathways (Fig. 9b). Estrogen also has a role in immunity, and enrichment of viral infection pathways like CMV, HTLV1 and HPV infections are also detected (Fig. 9b).

**Intricate dance of insulin and estrogen signaling and insulin resistance**. Some of the genes in the human samples were related to both insulin and estrogen signaling as well as insulin resistance including *IGFBP1, FGF23, WNT5A, CRP, ESR1, EGFR, FST, NCOR1, CD36, CYP17A1, SHBG, KISS1, LEPR*. The leptin receptor (*LEPR*) and its signaling are crucial for metabolism and obesity[31]. *LEPR* was downregulated upon return from spaceflight, regardless of the time interval (Figs. 7b, 8b, and 9b). The

*CYP17A1* gene was upregulated upon return (Fig. 7a). *CYP17A1*, encodes steroidogenic enzymes, 17 alpha-hydroxylase, and 17,20 lyase. These enzymes are responsible for the production of cortisol, estrogen and androgens, and their deficiencies have been linked to both hormonal imbalance and metabolic dysfunction such as hypertension. Most hormones bind to proteins in the circulation. Sex hormone binding globulin (SHBG) binds sex steroid hormones (estrogen and testosterone), and because of this interaction, the amount of SHBG is important as decreasing SHBG levels may lead to increasing free testosterone and estradiol levels, and vice versa. *SHGB* was upregulated 1 month upon return (R + 30d) and downregulated at the 3rd month (Figs. 7a, 8a, and 9a). Overlapping pathways were also observed such as PI3K-Akt signaling, cAMP signaling, insulin resistance, and estrogen signaling. It is well known that changes in estrogen signaling have been associated with insulin resistance[60], as seen in our results showing the enrichment of the estrogen signaling pathway via insulin resistance linked genes. Viral infection related pathways are enriched with both insulin and estrogen down-regulated genes (Figs. 7b and 9b).

**Insulin and estrogen signaling changes are observed in multiple human missions**. To explore if sex differently impacted insulin signaling, estrogen signaling, and insulin resistance, we utilized astronaut data (2 female and 2 male) from the first civilian commercial 3-day space mission, referred to as

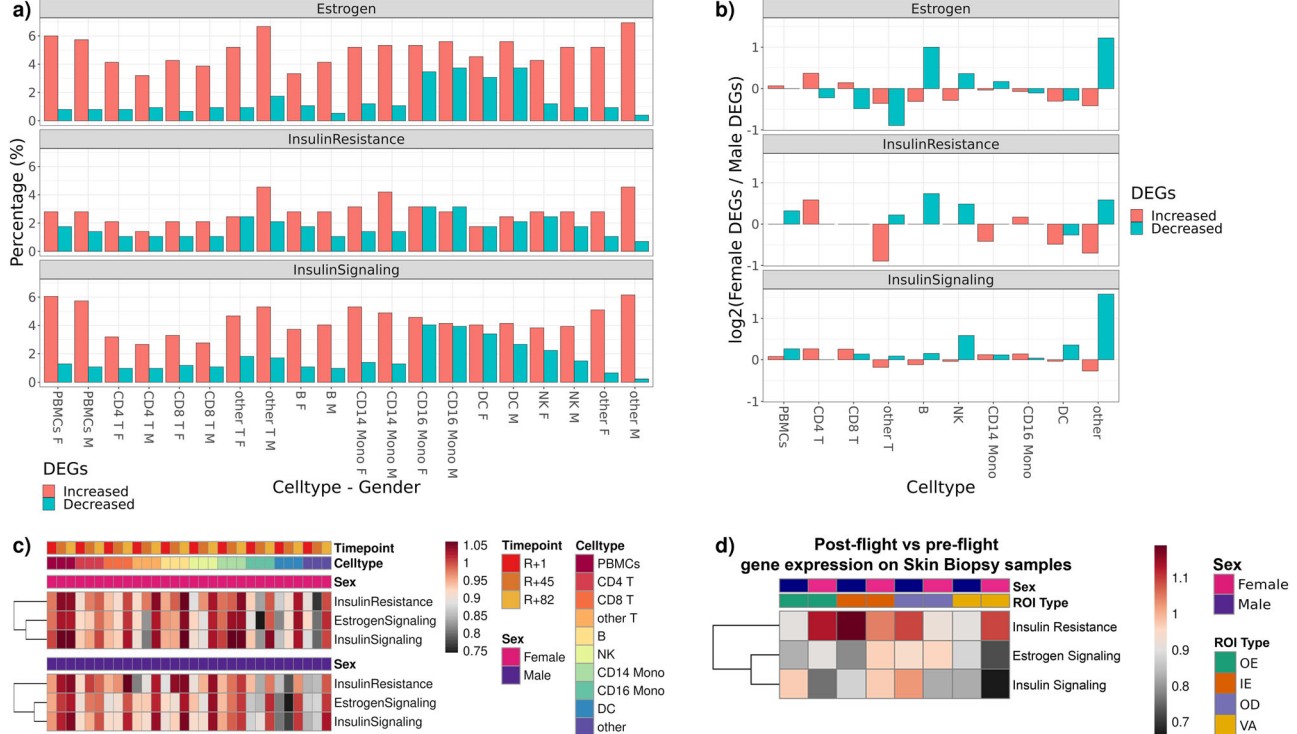

**Fig. 10 Insulin and estrogen signaling in inspiration4 astronaut PBMC and skin samples. a** Percent overlap of genes associated with insulin and estrogen signaling in single cell RNA-seq PBMC data, by cell types. **b** Log2 Female to male gene ratios from panel **c**. Fold changes of enrichment scores in response to spaceflight by sex, postflight collection time points, and cell types, PBMC data. The color scale bar represents fold change values are relative to preflight. **d** Fold changes of enrichment scores in postflight relative to preflight time point, by sex and tissue compartments (OE outer epidermis, IE inner epidermis, OD outer dermis, VA vasculature) in skin biopsy data. The color scale bar represents fold change values.

Inspiration4 (i4)[61]. From the i4 mission, single-cell gene expression data from peripheral blood mononuclear cells (PBMCs) and skin spatial transcriptomics data were generated.

Genes in the estrogen signaling, insulin signaling, and insulin resistance overlapped less than 10% with DEGs at R + 1 from PBMCs and subpopulations (CD4 T, CD8 T, other T, B, NK, CD14 Mono, CD16 Mono, DC) (Fig. 10a). Among these overlapping genes, the percentage of the upregulated genes were higher than downregulated genes (Fig. 10a). Log$_2$ Female DEGs to Male DEGs ratios varied across cell types, based on insulin and estrogen linked gene categories. Log$_2$ Female DEGs to Male DEGs ratios were positive in most cell types for insulin signaling (i.e. PBMCs, CD4 T, CD8 T, NK, DC, and 'other' subpopulations) (Fig. 10b). Certain cell types showed either a positive or negative ratio, depending on the hormone of interest. For instance, in females, in terms of DC, the DEGs ratio was positive for insulin-linked genes, but negative for estrogen-linked genes. In estrogen signaling, downregulated genes of B and other cell types were higher in females (Fig. 10b). In CD4 T cells, the ratios were positive in insulin, estrogen signaling and IR linked genes (Fig. 10b). We also calculated insulin resistance, estrogen signaling, and insulin signaling scores to track the changes over spaceflight. Higher enrichment scores were observed in all signaling pathways on R + 82, in comparison to R + 1 (Fig. 10c). In general, there was not a strong sex difference across cells (Fig. 10c). In certain cell types, differences in enrichment scores were observed depending on mission timeline and sex (Fig. 10c). In dendritic cells (DC), three pathways were positively enriched on R + 82 in both sexes; however, enrichment scores were different among females and males on R + 45 (Fig. 10c). T cell metabolism is modulated through insulin receptor signaling during inflammation and infection[62]. Alternatively, energy

consumption due to T cell activation drives upregulation of insulin receptors[63]. Therefore, it is evident that insulin signaling is a key immune-metabolic event during increased physiologic demands, as in i4 cohort findings. Similarly, estrogen receptor alpha is known to mediate T cell-dependent inflammatory reactions in a diseased state[12].

We also compared these findings in human skin data. To perform spatial transcriptomics analysis, we compared transcript profiles of preflight (L-44) and postflight (R + 1) tissue sections across four compartments within the skin, including outer epidermis (OE), inner epidermis (IE), outer dermis (OD) and vasculature (VA). While statistical analysis is limited by small sample sizes, we found slight differences between male and female skin compartments. Using the ssGSEA approach, overall, we observed increased enrichments in insulin resistance genes and decreased enrichments in estrogen and insulin signaling genes. There were slight sex differences, and the degrees of variation were region-specific. For example, we saw the strongest loss and sex-dependent variation of estrogen and insulin signaling signatures in VA regions (Fig. 10d). Insulin resistance linked genes were more enriched in females in the postflight vascular compartment and outer epidermis relative to preflight samples. In inner epidermis, both insulin and estrogen-signaling-linked genes showed larger changes in enrichments in females than in males.

## Discussion

Consistent with past reports that insulin signaling is a central regulator of gene expression changes and lipid metabolism in response to spaceflight in the worm, *C. elegans*[64,65], we find that gene expression changes related to insulin signaling occur in spaceflown rodents and humans. Further, we find that estrogen

signaling gene expression changes also occur in both spaceflown rodents and humans.

In mice, the liver and soleus muscle were the tissues that displayed the largest changes in global gene expression (Fig. 1b). These tissues also displayed large changes in insulin and estrogen signaling gene expression (Fig. 1c, d). In the liver, insulin receptor (INSR) and downstream kinases AKT1 and AKT2 were downregulated during spaceflight. These changes were accompanied by decreased expression of other insulin receptor signaling genes. The metabolic pathways directly controlled by insulin were inhibited, which resulted in activation of dysglycemia, insulin resistance, and hepatic steatosis. This last prediction is particularly striking given the increase in liver mass in these mice in a prior study utilizing the same cohort[66]. Hepatic steatosis is a direct consequence of insulin resistance and can in turn exacerbate insulin resistance, leading to the development of a metabolic syndrome, and thus resulting in a vicious cycle[67]. Prior studies have reported that during spaceflight, mice developed hepatomegaly[66] and hepatic lipid accumulation[68,69]. Further, the livers used in our analysis showed histologic evidence of lipid accumulation via Oil Red O staining in a prior study[68]. Thus, our results suggest that disrupted insulin signaling may underlie these changes. In addition to the insulin signaling changes, the impaired estrogen signaling we observed (z-score < −2), may contribute to hepatic insulin resistance.

In rodents, both male and female estrogen receptor knockout mice display insulin resistance[15]. Further, both insulin and estrogen actively play a role in metabolism and our findings also suggest that mitochondrial dysfunction, previously reported during spaceflight[2], may be due to alterations in insulin and estrogen signaling (Fig. 4a). PGC-1α, encoded by PPARGA1A, is a master regulator in metabolism and mitochondrial biogenesis[70]. PPARGC1A downregulation in the liver was observed in both insulin and estrogen gene expression changes. Additionally, estrogen regulates mitochondrial function and biogenesis[35–37]. In our liver data, downregulation of mitochondrial complex I, complex 2, and ATP synthase genes were observed in the estrogen signaling network (Fig. 4a). Thus, alterations in insulin and estrogen signaling in the liver may contribute to whole body alterations in metabolism in response to spaceflight.

Soleus muscle was the second most affected tissue in mouse studies. Soleus is a slow twitch (type I) muscle fiber, dense in mitochondria to provide sustained oxidative capacity for long-term demands. Prior studies show atrophy in the soleus muscle after spaceflight in both rodents and humans[66], and differential gene expressions between soleus and EDL muscle in rodents[71]. Myostatin (MSTN) controls muscle mass and has been reported to be upregulated in microgravity in the soleus muscle but not in EDL[72]; we confirm this observation. Other muscle atrophy related genes, FBX032 (Atrogin-1) and TRIM63, and mechanoreceptor PIEZO1 were also downregulated in the soleus. The downregulation of FBX032 and TRIM63 suggest the decline in soleus muscle size[66] is not caused by increases in muscle protein degradation. This is consistent with past reports that muscle protein degradation is not increased in flight, but rather that protein synthesis declines[73]. Previous studies in astronauts have suggested that altered insulin signaling may underlie muscle atrophy[25]. Further, in a prior study from the same mission, mice soleus muscle weight was decreased by 19%, despite no change in body weight and being physically active in space[66]. Given that insulin promotes protein synthesis, our finding of decreased expression in soleus of IRS2 (Insulin receptor substrate 2), one of major downstream insulin signaling, suggests that insulin alterations contribute to declines in protein synthesis to promote muscle atrophy in flight; this is also consistent with suggestions from spaceflown C. elegans[65]. On the contrary, insulin receptor

signaling was activated in the EDL (z-score: 2.52). Moreover, insulin-induced ACACA, a rate-limiting enzyme in lipogenesis, was upregulated indicating an insulin responsive state in the EDL. Further, the majority of the insulin signaling significant genes at the intersection of soleus and EDL were regulated in opposite directions, which shows distinctive insulin response in these muscles (Supplementary Data 1).

Estrogen also promotes muscle growth, regeneration, insulin sensitivity, and controls muscle inflammation[14,74]. Estrogen-related receptor beta and gamma (ESRRB, ESRRG) induce mitochondrial function in myotubes and have a role in shifting muscle fibers towards oxidative types[75]. Muscle-specific ESR1 knockout mice models showed decreased oxidative metabolism, impaired mitochondrial fission and glucose metabolism[76]. Thus, our findings that ESR1 receptor was upregulated and ESRRB and ESRRG were downregulated in soleus, while ESRRA, ESRRB, ESRRG were all upregulated in the EDL, suggest diverse actions of estrogen in muscle metabolism during spaceflight. Interestingly, while soleus size declined and EDL remained stable, as consistent with past studies[66], the gene expression changes in soleus and EDL were distinct with EDL displaying an enrichment for growth pathways (Fig. 3), also consistent with prior studies[71]. Estrogen receptor signaling pathways were activated in both EDL and soleus muscles. Soleus muscle mass recovery is reported to be slower after hindlimb suspension in ovariectomized mice[77]. Additionally, increased estrogen receptor expressions were reported following muscle trauma in regenerating myofibers[74]. Thus, estrogen receptor signaling activation and upregulation of ESR1 could be a compensatory mechanism against muscle disuse in the soleus muscle, whereas, in EDL the ESR signaling activation could support muscle growth and energy metabolism. Adipose tissue is another endocrine organ which is a part of crosstalk between muscle and liver. Our results are limited to muscle and liver, as adipose tissue was not available for analysis.

Studies in humans have more constraints than in model organisms, thus for the present study we were limited to blood samples and skin in spaceflown humans. However, as insulin and estrogen are circulating hormones, changes in their effects should be detected in these tissues; albeit at lower levels such as in the non-liver and non-soleus tissues in rodents (Fig. 1). As with the worm and rodent data, changes in insulin resistance related genes were detected inflight and postflight. Postflight samples from human subjects displayed upregulated insulin signaling genes (i.e., AKT1, RARRES2) as well as enriched insulin secretion and PI3K/AKT signaling (Fig. 8a), possibly suggesting decline in these pathways inflight. Additionally, postflight upregulation of the glucagon signaling pathway may indicate underlying postflight stress and may result in insulin resistance. Overall, postflight up- and down-regulated genes associated with insulin signaling displayed enriched lipid metabolism, hormones (i.e., estrogen, growth hormone, glucagon), circadian rhythm, and immune system pathways. Also, in the human data, we found the estrogen signaling pathway was enriched with insulin related genes. ESR1 was upregulated within 2 months (R + 3d, R + 30d, R + 60d) and then downregulated in the 3rd month (R + 120d). In correlation with this finding CYP17A1 was upregulated postflight. These findings may explain the enriched ovarian steroidogenesis and steroid hormone synthesis pathways seen postflight. Estrogen is closely related to the immune system, as an example, Treg cell distribution changes during menstrual cycles and increases during the follicular phase[78].

In our data, dendritic cells in females showed higher enrichment scores in estrogen and insulin linked genes on R + 45 than male counterparts (Fig. 10c). Sex difference was also observed in both rodent and human skin samples. In human skin, in the vascular region (VA) of the dermis, insulin resistance-related

genes were more enriched in females during postflight while insulin and estrogen signaling-related genes were less enriched (Fig. 10d). In rodent skin samples, certain genes associated with tissue remodeling and collagen synthesis (i.e. *COL1A1*, *COL6A3*, *COL6A1*, *COL5A1*) (Fig. 6b) were downregulated in male mice (Fig. 6b). Interestingly, increased urinary COL1A1 and COL3A1 levels have been reported in the NASA twin study[79]. Insulin and estrogen both play an active role in skin homeostasis, and insulin resistance has been linked to certain skin conditions[43]. Dermis is a connective tissue rich in vasculature, fibroblasts, collagen and elastic fibers, glands, hair follicles, and nerve endings. Both insulin and estrogen signaling exist in vascular endothelium and cause vasodilation via nitric oxide[80,81]. Insulin-resistant states could impair vascular endothelium[82,83]. Estrogen promotes skin and mucosal epithelialization and proliferation. Studies reported positive impact of estrogen in wound healing[84]. Clinically, topical estrogen is used in the treatment of vulvovaginal atrophy, and it proliferates epithelium[48]. Although our findings are limited by sample size, differences in the enrichment scores based on sex and skin compartments warrant future research on skin conditions after spaceflight in a sex and hormone-dependent manner. Further, due to our study setting, we cannot conclude on certain space-related environmental factors, such as lunar soil and dust, and their effects on insulin and estrogen signaling in the skin.

In conclusion, we find insulin and estrogen signaling are altered during and after spaceflight in rodents and people. These changes impact metabolism and forebode comorbidities. One advantage of the rodent data is that genes for pathways of interest were examined in various tissues from the same sample cohort, including mice of the same strain and age. However, a key limitation of our study is the small number of human participants. Future studies should include different rodent strains, more human subjects, and different mission profiles to gain more knowledge on these genes/pathways of interest and expand our results. Particular attention should be paid to the effects on reproductive hormone synthesis and actions, as these impact a wide range of tissues, including bone health, energy metabolism and, of course species survival via reproduction. Personalized healthcare using genomic data is the way of the future for both Earth and beyond[85]. Therefore, as we progress toward becoming an interplanetary species, further study on reproductive endocrinology and metabolism in space is essential for success.

## Methods

**Animal and sample collection**. All omics data related to mouse samples were obtained from NASA's GeneLab public omics repository[86]. The data derived from spaceflight experiments carried out onboard the ISS as part of the Rodent Research-1 (RR-1), RR-5, RR-7, and MHU-2 missions. A description of the experimental design, the corresponding type of tissues, and the data accessed are summarized in Supplementary Table 1. All previous animal experiments and methods were performed in accordance with the relevant guidelines at each principal investigators' institutions and were approved by the Institutional Animal Care and Use committee (IACUC). A detailed description of the methodology for obtaining tissue samples and processing is available in NASA's GeneLab platform.

For the RR-1 mission, in summary, forty female 12-week-old C57Bl/6 J (Jackson Lab, Bar Harbor, ME) were selected based on similar body weights for 4 experimental groups (16-week-old at launch, n = 10/group): baseline control, vivarium control, habitat control, and flight. Flight mice were launched onboard the ISS on 21 September 2014. Relevant to this study, the habitat control was placed in the ISS Environmental Simulator at NASA KSC on a 4-day delay to mimic flight temperature, $CO_2$, and humidity conditions on the ISS for the duration of spaceflight. Flight mice were exposed to microgravity for a total of 37 days (33 days on ISS and 4 days in the Dragon Capsule). Mice were euthanized by injection of Euthasol followed by cervical dislocation and immediately fast frozen intact or partially dissected prior to carcasses being frozen in the Minus Eighty Degree Laboratory Freezer (MELFI) aboard the ISS. All ground mice were processed similarly. Flight mice carcasses returned to Earth February 2015 and were then maintained along with the control mice frozen at the Biospecimen Sharing Program (BSP) at the Ames Research Center until dissection[66].

Samples were collected and processed as previously described[27,68]. Mouse carcasses were removed from −80 °C storage and thawed at room temperature for 15–20 min. Multiple tissues were collected from each carcass. This study was based on the following tissue types, the Adrenal glands (OSD-98), Extensor digitorum longus muscle (OSD-99), Eye (OSD-100), Gastrocnemius muscle (OSD-101), Kidney (OSD-102), Quadriceps Muscle (OSD-103), Soleus muscle (OSD-104), Tibialis anterior muscle (OSD-105), and Liver (OSD-168, OSD-48).

**RNA sequencing and analysis**. Protocol for each dataset is available on the GeneLab platform (Table S1). Data validation and quality control was performed with FASTQC and Trim Galore! read alignment to the mouse genome using STAR RNA-seq aligner generation of gene-level expected count data with RSEM[87]. UpSet plots were generated using the UpSetR package in R. The UpSet plot bars were sorted from the largest number of genes to the smallest number of genes in each intersection set with each set having ≥ 2 genes. Heat maps were created using packages available through R (pheatmap and complexheatmap). Hierarchical clustering was used for the heatmaps where Euclidean distance was used as a distance measure and the complete linkage method was used to find similar clusters.

For the gene set enrichment analysis (GSEA) Pathway Analysis, the Genelab datasets are re-analyzed to produce the t-score. The Differential Expression (DE) analysis is accomplished using DESeq2 version 1.26.0[88], in R software version 4.1.2. Expected counts from the RSEM step are extracted and rounded up to the next integer and used as input for DE analysis. The ensembl IDs of the genes in the DE datasets were annotated to their corresponding gene symbols using the R package biomaRt[89,90]. The murine skin datasets were analyzed via a similar pipeline available at https://github.com/henrycope/spaceflight-skin-transcriptomics-insulin-estrogen, using R version 4.1.0, DESeq2 version 1.32, and biomaRt version 2.48.3.

We performed GSEA on the differentially expressed datasets using fGSEA[91] to determine to what extent insulin and estrogen signaling pathways were impacted in spaceflight. Estrogen- and insulin- specific gene pathways were selected using the Molecular Signatures Database (MSigDB)[91,92]. The latest version MSigDB v2022.1 was used in the pathway search. The keyword "estrogen" was used to identify the estrogen-specific pathways through all database collections and all contributors within the *Mus musculus* species. Similarly, the keyword "insulin" was used to identify insulin-specific pathways through all contributors and *Mus musculus* database collections. The outcome of the fGSEA analysis is the normalized enrichment score (NES) of any given pathway, alongside valuable information on the analysis, such as the adjusted p-value, and the leading-edge genes of the pathway. The NES is the enrichment score normalized based on the number of genes in the gene set. It indicates the representation of the pathway genes in the dataset gene list, which is priorly ranked according to the values of the t-score of the genes. A positive NES indicates that the pathway genes are mostly represented at the top of the gene list, while a negative NES indicates that the pathway genes are mostly represented at the bottom of the gene list. The plots of normalized enrichment score (NES) of the different pathways for a given dataset are rendered using ggplot2[93], and the heatmaps across different datasets or different genes are rendered using the ComplexHeatmap R package.

We used Ingenuity Pathway Analysis (IPA) software (Ingenuity Systems) for canonical pathway analysis and subsequent predictions in each tissue with statistically significant genes with a fold-change ≥ 1.2 (and ≤ −1.2) comparing flight conditions versus habitat ground controls (FDR < 0.05 as stated above). Low fold-change has become quite standard when trying to identify genes that are differentially expressed, as low cutoffs are less affected by different data normalization schemes and they are less likely to eliminate key genes operating under very tight level regulation[61]. IPA was used to predict statistically significant activation or inhibition of canonical pathways using activation Z-score statistics (≥2, activated or ≤−2, inhibited)[94].

**Cell-Free Epigenome JAXA Study plasma cfRNA data**. Aggregated normalized RNA expression values were shared through NASA's GeneLab platform with accession number: OSD-530[51]. Blood samples were collected from 6 astronauts before, during, and after the spaceflight on the ISS. cfRNA samples were obtained by purifying total RNA from plasma samples and analyzed by RNA-seq. Mean expression values from normalized read counts of 6 astronauts were used for the analysis in this report. Heatmaps were made for the specific genes on the normalized values per time point using R package pheatmap version 1.0.12.

**Inspiration 4 (i4) astronaut sample collection**. Inspiration4 was the world's first all-civilian mission to orbit Earth. Four civilians, two males and two females, spent three days in Low-Earth Orbit (LEO) at 585 km above Earth. The mission launched from NASA Kennedy Space Center on 15 September 2021 and splashed down in the Atlantic Ocean near Cape Canaveral on September 18th, 2021. Several human-related experiments were carried out to study the effects of spaceflight on human health and performance in collaboration with SpaceX, the Translational Research Institute for Space Health (TRISH) at Baylor College of Medicine (BCM), and Weill Cornell Medicine. The experiments carried out on the Inspiration4 crewmembers were performed in accordance with the relevant guidelines at the principal investigators' institutions. Moreover, the different study designs and the

corresponding methods to collect and analyze the biological samples were approved by BCM IRB. All biological data derived from the Inspiration4 mission were collected pre and post flights. For this study, only data from blood samples and skin biopsies were used. Pre-flight samples were collected at L-92, L-44, and L-3 days prior to launch to space. Upon return, post-flight samples were collected at R + 1, R + 45, and R + 82 days.

**i4 PBMC Single-cell sequencing and analysis.** Blood samples were collected before (Pre-launch: L-92, L-44, and L-3) and after (Return; R + 1, R + 45, and R + 82) the spaceflight. Chromium Next GEM Single Cell 5' v2, 10x Genomics was used to generate single cell data from isolated PBMCs. Subpopulations were annotated based on Azimuth human PBMC ref. [95]. ssGSEA was used to calculate the pathway scores of Insulin Resistance, Estrogen Signaling, Insulin Signaling. Heatmaps are generated by the pheatmap R package.

**Skin spatial transcriptomics.** For skin spatial transcriptomics data, 4 mm diameter skin biopsies were obtained from all Inspiration4 crew members, once before flight and as soon as possible after return (L-44 and R + 1). These biopsies were flash frozen and processed with the NanoString GeoMx platform. A 20x scan was used to select 95 freeform regions of interest (ROIs) to guide selection of regions of interest. From immunofluorescence staining, we identified epidermis, dermis (OD), and vasculatures (VA); we subdivided epidermis into outer and inner layers, which corresponded to granular and spinous layer for outer epidermal (OE), and basal layer for inner epidermal (IE). GeoMx WTA sequencing reads from NovaSeq6000 were compiled into FASTQ files corresponding to each ROI and converted to digital count conversion files using the NanoString GeoMx NGS DnD Pipeline. From the Q3 normalized count matrix that accounts for factors such as capture area, cellularity, and read quality, the DESeq2 method was used to perform differential expression analysis.

**Proteomics sample preparation and data processing.** The specific details of the proteomics method was previously published and can be found on GeneLab's platform. In brief, after protein homogenization, solubilization, and hydrolysis, resulting peptides were dried and labeled using TMT isobaric tags. The labeled peptides were analyzed on the Orbitrap Fusion mass spectrometer (Thermo Fisher Scientific). The Fusion MS instrument was operated in positive ion data-dependent mode with synchronous precursor selection (SPS)-MS3 analysis for reporter ion quantitation. Peptide and protein identification and quantification was performed with MaxQuant (1.6.2.3) using a mouse reference database from Uniprot. Protein groups containing matches to decoys or contaminants were discarded. Subsequently, Internal Reference Scaling (IRS) method was employed to normalize protein intensities between different TMT experiments using common proteins in pooled internal standards. The data was log2 transformed and scaled by subtracting the median for each sample. Limma was employed to determine differentially protein abundance between the flight group versus the ground control. All data was plotted as heatmaps using pheatmap (ver.1.0.12).

**Statistics and reproducibility.** All statistics are described throughout the methods. Briefly, for all RNA-seq data we utilized a cutoff of FDR (or adjusted $p$-value) < 0.05. For pathway analysis (i.e. GSEA analysis) as indicated in the methods we used a FDR < 0.25, which is the recommended statistical cutoff to utilize for GSEA analysis[92].

**Reporting summary.** Further information on research design is available in the Nature Portfolio Reporting Summary linked to this article.

## Data availability

Majority of the datasets utilized are publicly available at the NASA GeneLab website (https://osdr.nasa.gov/bio/repo), including rodent data; Adrenal glands (OSD-98), Extensor digitorum longus muscle (OSD-99), Eye (OSD-100), Gastrocnemius muscle (OSD-101), Kidney (OSD-102), Quadriceps Muscle (OSD-103), Soleus muscle (OSD-104), Tibialis anterior muscle (OSD-105), Liver (OSD-168, OSD-48), Skin (OSD-238, OSD-239, OSD-240, OSD-241, OSD-254), JAXA CFE data (OSD-530).

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

## Acknowledgements

N.S. was supported by grants from NASA [NSSC22K0250; NSSC22K0278] and acknowledges the support of the Osteopathic Heritage Foundation through funding for the Osteopathic Heritage Foundation Ralph S. Licklider, D.O., Research Endowment in the Heritage College of Osteopathic Medicine. A.B. was supported by NASA grant 16-ROSBFP_GL-0005: NNH16ZTT001N-FG Appendix G: Solicitation of Proposals for Flight and Ground Space Biology Research (Award Number: 80NSSC19K0883). C.E.M. thanks the Scientific Computing Unit (SCU) at WCM, the WorldQuant and GI Research Foundation, NASA (NNX14AH50G, NNX17AB26G, NNH18ZTT001N-FG2, 80NSSC22K0254, 80NSSC23K0832, the Translational Research Institute through NASA Cooperative Agreement NNX16AO69A), the National Institutes of Health (R01MH117406), and the LLS (MCL7001-18, LLS 9238-16, 7029-23). H.C. is supported by the Horizon Center for Doctoral Training at the University of Nottingham (UKRI grant no. EP/S023305/1). Graphical abstract (Fig. 1a) Created with BioRender.com.

## Author contributions

Conceptualization: A.B. and B.A.M.; methodology: A.B., B.A.M., M.T., and R.K.; formal analysis: A.B., B.A.M., M.T., R.K., V.Z., J.K., H.C., H.F., J.P., R.S.S., and F.K. Investigation: A.B., C.E.M., F.K., B.A.M. M.M., R.M.S., J.K., M.T., V.Z., and R.K.; i4 data and omics: C.E.M., E.O., J.P., M.Y., R.V.K., and J.K.; JAXA data: M.M.; resource: A.B., C.E.M., and M.M.; original draft: B.A.M., A.B., N.S., A.T., J.K., V.Z., A.R.I.,M.T., R.K. and R.M.S.; review and editing: Y.K.C., D.P., J.B., S.L.Y., M.T., R.K., V.Z., J.K., H.C., H.F., J.P., R.S.S., F.K., C.E.M., N.S., H.F., V.Z., J.K., M.M., R.M.S., B.A.M., and A.B.; figures and visualization: A.B., M.T., J.K., R.K, B.A.M, A.T., N.S., H.C., H.F., V.Z., J.P., and R.S.S.; funding acquisition: A.B. and C.E.M. (for i4 study); supervision: A.B. and B.A.M.

## Competing interests

The authors declare no competing interests.
