## [Peer Review File · Communications Biology]

REVIEWERS' COMMENTS:

Reviewer #1 (Remarks to the Author):

The authors have made an effort to answer some of the questions raised. They have added additional information (publicly available proteomic data) to support some of their initial conclusion.

The author claims that they could not get human liver biopsies (which is acceptable), but many serum levels are good indicators of liver metabolism status. This information could have been easily added in the current manuscript.

One major issue I would see before acceptance to publication in Commsbiology is to clarify gene and protein expression data.

Nomenclature generally follows the conventions of human nomenclature. Gene symbols generally are italicized, with all letters in uppercase (e.g., GLUT4, for Glucose transporter 4). Protein designations are the same as the gene symbol but are not italicized; all letters are in uppercase (GLUT4).

The authors should revise the current version and clarify to which dataset they refer to, in order to help the reader.

Moreover, they are still a lot of typo errors in the manuscript. The authors should carefully read it to correct them before final submission.

Reviewer #2 (Remarks to the Author):

The authors have answered to reviewers' concerns.

Reviewer #3 (Remarks to the Author):

Thank you to the authors for addressing the concerns raised and revising the manuscript. In particular, the reviewer appreciates the changes that describe the caveats to the study and the addition of the transcriptomic data on skin from mice flown to the ISS presented in Figure 6. However, it is concerning that this data did not show consistent findings across different strains of mice, length of time in space, tissue type (dorsal vs femoral skin) and diet. Thus, it is difficult to make any conclusions that spaceflight can alter insulin and estrogen signaling in skin. One could argue that the lack of consistency shows that spaceflight does not have an effect. Nevertheless, the reviewer feels that the authors have adequately described the data and have not overstated the findings.

Dear Reviewers,

We thank the reviewers for the comments. We believe based on these final revisions we have made that this manuscript is now stronger and much improved. We have addressed the comments by the reviewers and our responses appear in red font below the original reviewer comment. We look forward to the next steps.

Afshin Beheshti, PhD

REVIEWERS' COMMENTS:

Reviewer #1 (Remarks to the Author):

The authors have made an effort to answer some of the questions raised. They have added additional information (publicly available proteomic data) to support some of their initial conclusion.

The author claims that they could not get human liver biopsies (which is acceptable), but many serum levels are good indicators of liver metabolism status. This information could have been easily added in the current manuscript.

One major issue I would see before acceptance to publication in Commsbiology is to clarify gene and protein expression data.

Nomenclature generally follows the conventions of human nomenclature. Gene symbols generally are italicized, with all letters in uppercase (e.g., GLUT4, for Glucose transporter 4). Protein designations are the same as the gene symbol but are not italicized; all letters are in uppercase (GLUT4).

We thank the reviewer for this suggestion and have made the changes as suggested with the proper nomenclature throughout the manuscript.

The authors should revise the current version and clarify to which dataset they refer to, in order to help the reader.

We have provided the dataset IDs throughout the manuscript next the tissues as the reviewer has suggested.

Moreover, they are still a lot of typo errors in the manuscript. The authors should carefully read it to correct them before final submission.

We have also gone through the manuscript and correct all typos and errors.

Reviewer #2 (Remarks to the Author):

The authors have answered to reviewers' concerns.

We thank the reviewer for their comments and appreciate all their input.

Reviewer #3 (Remarks to the Author):

Thank you to the authors for addressing the concerns raised and revising the manuscript. In particular, the reviewer appreciates the changes that describe the caveats to the study and the addition of the transcriptomic data on skin from mice flown to the ISS presented in Figure 6. However, it is concerning that this data did not show consistent findings across different strains of mice, length of time in space, tissue type (dorsal vs femoral skin) and diet. Thus, it is difficult to make any conclusions that spaceflight can alter insulin and estrogen signaling in skin. One could argue that the lack of consistency shows that spaceflight does not have an effect. Nevertheless, the reviewer feels that the authors have adequately described the data and have not overstated the findings.

We thank the reviewer for their comments and appreciate all their input.